# Antinociceptive and Cardiorespiratory Effects of a Single Dose of Dexmedetomidine in Laboratory Mice Subjected to Craniotomy under General Anaesthesia with Isoflurane and Carprofen or Meloxicam

**DOI:** 10.3390/ani14060913

**Published:** 2024-03-15

**Authors:** Anika R. Schiele, Inken S. Henze, Regula Bettschart-Wolfensberger, Thomas C. Gent

**Affiliations:** Anaesthesiology Section, Department of Clinical Diagnostics and Services, Vetsuisse Faculty, University of Zurich, 8057 Zurich, Switzerland; ihenze@vetclinics.uzh.ch (I.S.H.); rbettschart@vetclinics.uzh.ch (R.B.-W.)

**Keywords:** mice, craniotomy, dexmedetomidine, analgesia, anti-nociception, anaesthesia, animal welfare

## Abstract

**Simple Summary:**

Craniotomy is a commonly performed surgical procedure in neuroscientific research that causes substantial pain in laboratory animals. The animals’ invidious situation has received little previous attention, and therefore, to respect the 3R guidelines, including the minimisation of the animals’ pain and suffering, we aimed to improve the conventional anaesthetic and analgesic regimens. The anti-nociceptive effects of four different drug combinations in laboratory mice undergoing general anaesthesia for craniotomies were investigated. Specifically, the efficacy of dexmedetomidine, which is known to have analgesic, sedative, and cardiorespiratory stabilising properties in other species, was evaluated. Our study demonstrated the beneficial effects of dexmedetomidine in mice as the signs of nociception were mitigated in the respective treatment groups.

**Abstract:**

Pain refinement represents an important aspect of animal welfare in laboratory animals. Refining analgesia regimens in mice undergoing craniotomy has been sparsely investigated. Here, we sought to investigate the effect of dexmedetomidine in combination with other analgesic drugs on intraoperative anti-nociceptive effects and cardiorespiratory stability. All mice were anaesthetised with isoflurane and received local lidocaine infiltration at the surgical site. Mice were randomised into treatment groups consisting of either carprofen 5 mg kg^−1^ or meloxicam 5 mg kg^−1^ with or without dexmedetomidine 0.1 mg kg^−1^ administered subcutaneously. Intra-anaesthetic heart rates, breathing rates, isoflurane requirements, and arterial oxygen saturations were continuously monitored. We found that administration of dexmedetomidine significantly improved heart and breathing rate stability during two of four noxious stimuli (skin incision and whisker stimulation) compared to non-dexmedetomidine-treated mice and lowered isoflurane requirements throughout anaesthesia by 5–6%. No significant differences were found between carprofen and meloxicam. These results demonstrate that dexmedetomidine reduces nociception and provides intra-anaesthetic haemodynamic and respiratory stability in mice. In conclusion, the addition of dexmedetomidine to anaesthetic regimes for craniotomy offers a refinement over current practice for laboratory mice.

## 1. Introduction

Craniotomy is an invasive procedure to access the brain for surgical interventions and thus enable techniques, which are commonly performed in neuroscientific research, such as intracranial injection for gene delivery [1,2], optical implant [2], cannulation [3], or electrical stimulation [4]. The number of craniotomies performed in laboratory animals, especially rodents, has increased progressively over time and is currently the most common neuroscientific surgical procedure in mice [5].

Craniotomy is a well-documented cause of significant perioperative morbidity and mortality in humans, with patients experiencing moderate to severe intraoperative and postoperative pain [6]. However, there is a distinct lack of data investigating perioperative pain and its management in mice undergoing craniotomy. Human [7] and animal [8] studies show that perioperative morbidity such as pain can alter the physiologic responses of organ systems, behaviour, and immune status, which may lead to variation in experimental results, impairing both the reliability and repeatability of scientific studies [8,9]. Furthermore, refinement of animal experiments is part of the 3R’s principles and has to be implemented in modern research practices according to a directive in the European Union (EU) [5]. The alleviation of pain is essential to the reproduction and reduced variability of scientific results as well as the minimisation of the impact of neuroscientific procedures on the life of laboratory rodents undergoing craniotomies [8]. However, recent data suggests that 36 percent (%) of experimental procedures in laboratory mice are conducted with the use of pure inhalation anaesthesia, which does not provide any analgesia [5]. To the authors knowledge, there are no evidence-based recommendations for the optimisation of intraoperative analgesia and anaesthetic stability during craniotomies performed under isoflurane anaesthesia in laboratory rodents. 

In humans undergoing craniotomy, intraoperative constant-rate infusion of dexmedetomidine reduced postoperative pain, the consumption of other analgesic agents, and frequently occurring complications like nausea and vomiting [10] as well as hypertension and tachycardia [11]. In different animal species, dexmedetomidine significantly reduces the minimal alveolar concentration (MAC) of inhalant anaesthetics [12,13,14] and the minimum infusion rate of anaesthetics used for total intravenous anaesthesia [15]. Beside the intraoperative administration, dexmedetomidine in humans is frequently utilised for neurointensive care as it allows analgesia without respiratory depression and provides neuroprotection and haemodynamic stability [16,17]. In mice, dexmedetomidine is not only neuroprotective [18,19,20] but also has analgesic effects [21,22], which are dose-dependent [21,23]. Therefore, we hypothesised that dexmedetomidine may improve intraoperative cardiorespiratory stability as well as perioperative analgesia in mice undergoing craniotomy and would provide superior analgesia compared to meloxicam or carprofen. We investigated the effects of dexmedetomidine and other analgesic drugs, to recommend for mice undergoing craniotomy drugs that assure intraoperative haemodynamic stability and analgesia.

## 2. Materials and Methods

### 2.1. Ethical Approval

This prospective, randomised, and blinded study was approved by the Canton of Zürich veterinary office (license number 002/2020).

Additional data collected from animals in this study are reported elsewhere.

### 2.2. Animals

Adult female (*n* = 20; 76 (62–183) days old; 21.0 (18.3–31.2) g body weight) and male (*n* = 21; 82 (53–220) days old; 28.9 (23.4–44.5) g body weight) C57BL/6J mice (Charles Rivers Laboratories, Cologne, Germany) were included in the study. Only animals that were considered healthy, showed normal behaviour, and showed normal food and water intake were included in the study. Animals that showed a hunched back, piloerection, inappetence, or lethargy were excluded.

### 2.3. Housing

Animals were housed on a 12/12 light cycle (lights on at 08:00) in exhaust individual ventilated cages (EIVC) (39.8 × 19.4 × 18.1 cm; NexGen 500, Allentown LLC., Allentown, NJ, USA) at a temperature of 20–25 °C and humidity of 35–75% as same-sex litter mates in groups of three. Bedding consisted of wood chips (Safe Select, Safe Lab, Augy, France), a piece of paper tissue serving as nesting material, and a resting box. Food (Granovit 3436, Granovit AG, Kaiseraugst, Switzerland) and water were available ad libitum. The animals were transferred to clean cages by animal technicians every seven days.

### 2.4. Anaesthesia Induction and Treatments

The timeline for the experiments is summarised in Figure 1a.

Mice were randomly assigned to one of four treatment groups (carprofen 5 mg kg^−1^ only (Group C), meloxicam 5 mg kg^−1^ only (Group M), carprofen 5 mg kg^−1^ and dexmedetomidine 0.1 mg kg^−1^ (Group CD), and meloxicam 5 mg kg^−1^ and dexmedetomidine 0.1 mg kg^−1^ (Group MD)). Randomisation was performed by drawing lots from two envelopes that each contained as many lots as mice, one envelope each with 20 lots for male and female animals. Each envelope contained an equal number of four different lots (one per group) marked with the relevant group’s short name. Randomisation was performed by an author who also prepared the treatments but did not perform the procedures. The procedures were performed by two researchers who were blinded to the treatments. Anaesthesia induction was performed in all animals in a plexiglass induction chamber (Figure 2; 28.6 × 17.8 × 15.2 cm, EZ-1785 Sure-Seal Large Mouse/Rat Chamber, E-Z Systems Inc., Palmer, PA, USA) with 3.0% isoflurane (IsoFlo^®^, Zoetis GmbH, Delémont, Switzerland) in 100% oxygen at a flow rate of 1.0 L min^−1^. Isoflurane concentration was reduced to 2.0% after 3 min. Anaesthesia was performed by one of the researchers while the second one performed the surgical procedure and optical imaging. 

They were weighed on an electronic scale (OHAUS Europe GmbH, Greifensee, Switzerland) and transferred to the preparation area. Here, all animals were put on a non-feedback heating mat (Tonkey Electrical Technology Co., Ltd., Shenzhen, China) with a set point of 40 °C and positioned with the nose in a gas delivery mask for continued isoflurane delivery. Vitamin A eye ointment (Vitamin A <Blache> Augensalbe, Bausch & Lomb Swiss AG, Zug, Switzerland) was applied liberally to both eyes.

Drugs (Rimadyl 50 mg mL^−1^, Zoetis GmbH, Switzerland; Metacam 5 mg mL^−1^, Boeringer Ingelheim GmbH, Basel, Switzerland; Dexdomitor 0.5 mg mL^−1^, Orion Pharma, Espoo, Finland) were drawn up and mixed in case of the addition of dexmedetomidine. Then, they were diluted to a total volume of 5 mL kg^−1^ for each mouse with sterile saline (NaCl 0.9%, Bichsel AG, Interlaken, Switzerland) and administered subcutaneously via an insulin syringe (BD Micro-Fine™+, Becton Dickinson France SAS, Le Pont-de-Claix, France) with a 30-gauge needle (Omnican^®^, B. Braun, Sempach, Switzerland).

### 2.5. Anaesthesia Monitoring

Anaesthesia monitoring was conducted in the surgical area and consisted of continuous pulse oximetry, capnography, and temperature measurement. A pulse oximeter probe (MouseOx^®^ Plus, Starr Life Sciences Corporation, Oakmont, PA, USA) was placed on the right thigh, recording pulse rate, breathing rate, and oxygen saturation. Capnography and respiratory gas measurement via a multiparameter monitor (GE Carescape™ B450, Anandic Medical Systems AG, Feuerthalen, Switzerland) with the sampling tube connected to an empty catheter (SurflashTM I.V. Catheter 24 G, 0.67 × 19 mm, yellow, Terumo Deutschland GmbH, Spreitenbach, Switzerland) and placed directly in front of the mice’s nares enabled measurement of end tidal CO_2_, F_i_O_2,_ end tidal isoflurane, and breathing rate to have another monitoring source next to the MouseOx^®^ Plus due to the lack of reliability as described below. For temperature measurement in the surgical area, a rectal probe was placed, which provided feedback to a heating mat (Homeothermic Monitoring System, Harvard Apparatus, Cambridge, MA, USA) set to regulate body temperature at 37.0 °C. Animals were covered with a sterile drape to save heat.

End tidal isoflurane, vaporiser setting, F_i_O_2_, SpO_2_, body temperature, heart rate, and breathing rate were recorded every 5 min. Simultaneously, the MouseOx^®^ Plus recorded the parameters described above continuously every second. The files were exported as a text file at the end of experimentation and converted to Excel files for further analysis.

### 2.6. Surgical Procedures

A schematic overview of all surgical procedures is represented in Figure 1b.

The hair was shaved from the scalp and from both the medial and lateral aspects of the right thigh using a battery powered clipper. Hair removal cream (Bikini & Achseln Haarentfernungs-Creme, Veet, Reckitt Benckiser AG, Wallisellen, Switzerland) was applied to the scalp and left for 5 min before being removed with a cotton bud. Thereafter, the scalp was aseptically prepared with two separate washes with 70% ethanol (Softasept^®^ N, B. Braun, Sempach, Switzerland) using a cotton bud.

Animals were then transferred to the surgical area, where they were placed in a stereotaxic frame (Model 940, David Kopf Instruments, Tujunga, CA, USA) secured by 60° non-rupture ear bars (Model 922, David Kopf Instruments, Tujunga, CA, USA) and a mouse incisor bar with an integrated anaesthesia gas mask (Model 923-B, David Kopf Instruments, Tujunga, CA, USA). Anaesthesia was maintained by isoflurane in oxygen delivered by the integrated anaesthesia gas mask connected to an anaesthesia machine (VetFlo™ Vaporiser Single Channel Anaesthesia System, Kent Scientific Corporation, Torrington, CN, USA), continued with a vaporiser setting of 2.0% in 0.5 L min^−1^ of 100% oxygen. The scalp was then cleaned again with 70% ethanol. Lidocaine, 2 mg kg^−1^ (Lidocain HCl “Bichsel” 20 mg mL^−1^, Bichsel AG, Interlaken, Switzerland), was injected subcutaneously into the scalp using an insulin syringe with a 30-gauge needle. Isoflurane concentration was adjusted to maintain a sufficient plane of anaesthesia according to anaesthetic depth evaluated through respiratory rate, breathing pattern, and reaction to surgical or auditory stimulation. Anaesthetic depth was considered adequate when the unstimulated animals showed a respiratory rate between 68 and 80 breaths per minute. Isoflurane concentration was increased by 0.2% at respiratory rates above and decreased at respiratory rates below this range. Under general anaesthesia, the costoabdominal breathing pattern of the mice was considered normal. When a change to an inverse breathing pattern was observed, the isoflurane concentration was reduced by 0.2%. When showing a reaction to surgical or auditory stimulation in the form of increased respiratory rates for a longer than 30 s period, the isoflurane concentration was increased by 0.2%. Ten minutes after subcutaneous administration of lidocaine, a single incision from the level of the lateral canthi of the eyes to the nuchal crest was performed using a No. 15 scalpel blade. The skin was held apart using stay sutures (one per side; Prolene 5-0, Johnson & Johnson AG, Zug, Switzerland), and the subcutaneous tissue was removed from the skull using a sterile cotton bud. The skull was then levelled using a sterile probe mounted in a stereotaxic holder (Model 1776, David Kopf Instruments, Tujunga, CA, USA) using bregma and lambda as datum points.

Intrinsic optical imaging of the left barrel cortex was performed by piezoelectric stimulation of whiskers and by recording the neural activity over 10 min using a charge-coupled device camera (CCD). This was repeated five times. Craniotomy was performed at AP: −1.65, ML—0.5 mL with respect to bregma, using a 0.4 mm burr in a precision micro-drill (Foredom Electric Co., Bethel, CT, USA) rotating at 30,000 rpm. The injection needle was lowered into the hole to a depth between 3.5 and 4.0 mm ventral to bregma at a 20° angle to target the central thalamus. A virus (HSV-hEF1a-ChR2(H134R)-mCherry); RN408; Massachusetts General Hospital, Gene Delivery Technology Core) was injected (200 nL, 100 nL min^−1^) via a 33-gauge stainless steel needle connected via PVC tubing to a 10 µL Hamilton syringe, mounted in a precision syringe driver (World Precision Instruments, Friedberg, Germany). The needle was left in situ for 10 min following injection to allow for diffusion, and the needle was then removed. The skull was rinsed with 2 mL sterile Ringer’s Lactate solution at body temperature and cleaned with a sterile cotton tip. The stay sutures were removed, and the skin incision was closed using two cruciate sutures (5-0 Prolene). 

### 2.7. Postoperative Management and Observation

After surgery, isoflurane delivery was discontinued, and animals were placed and recovered individually in open-top cages (transparent, open-top cages, 35.1 × 20.1 × 29.8 cm) on top of a non-feedback warming mat for postoperative monitoring. Thereafter, to avoid stress, three cages, each containing one mouse, were placed next to each other under the same conditions as described before (see Section 2.3).

Postoperative analgesia consisted of the same NSAID as used intraoperatively, i.e., meloxicam 5 mg kg^−1^ or carprofen 5 mg kg^−1^. The drug was administered subcutaneously once every 24 h for five days in the morning. After another two days of surveillance in the open-top cages, animals were transferred to the EIVC cages (see Section 2.3) and brought back to the standard husbandry room. Animals were weighed and scored twice daily for one week. Scoring included four categories: weight loss, alterations in behaviour including eating, drinking, defecating, and nesting behaviour, alterations of gait and posture, and abnormal breathing frequency and pattern. Each category was graded with 0, 1, or 2, meaning normal, mildly altered, or severely altered. An individual score of 2 or more, as well as a total score of 3 or more, was taken as the threshold for rescue analgesia consisting of buprenorphine 0.1 mg kg^−1^ subcutaneously. At 30 min after administration, animals were reassessed and, according to the described criteria, received another injection of rescue analgesia or not. In case of continued weight loss over 5% of the initial body weight, sugar-rich food was offered (from day one after surgery on) and a crystalloid fluid bolus of 10 mL kg^−1^ was administered (from two days after surgery on). Termination criteria included continued weight loss of 15% or more from the initial body weight, a total score of 5 or more, or an individual score of 2 in two categories, hyper aggression on handling, immobility over two hours, non-healing skin incisions after three days, and no improvement from two consecutive injections of rescue analgesia.

### 2.8. Statistical Analysis

Data were analysed by one-way ANOVA or Kruskal–Wallis test in GraphPad Prism (version 6.2.1) or Microsoft Excel (Microsoft 365 for Enterprise, Washington, DC, USA). The sample size was calculated using G*Power 3.1.9.7 (Heinrich-Heine-Universität Düsseldorf, Düsseldorf, Germany). The total sample size was 24 animals, considering an error probability of 0.05, a confidence level of 95%, a power (1-b probability of error) of 0.90, and a correction between repeated measures of 0.5 for four experimental groups. Residuals were tested for normal distribution with the D’Agostino–Pearson test and for heteroscedasticity by Brown–Forsythe test; if both tests were passed, one-way ANOVA followed by Tukey’s multiple comparison test was used; otherwise, data were analysed with the Kruskal–Wallis test followed by Dunn’s multiple comparison test. Significance was set at adjusted *p* < 0.05. Values are reported as mean ± S.E.M. when ANOVA was used and as median (range) when Kruskal–Wallis test was used. Comparisons are to control conditions, unless otherwise stated.

To investigate the potential effects of different drugs at several events, the time intervals from drug administration to the respective event were determined by calculating the mean (range) values in Microsoft Excel (Version 2402, Microsoft 365 for Enterprise).

The mean (range) induction time was calculated using Microsoft Excel (Microsoft 365 for Enterprise), whereas the duration was determined by analysing the respective video tape.

## 3. Results

One mouse was excluded due to the suspicion of an incorrectly prepared drug syringe. One mouse (group C) reached termination criteria (weight loss over 15%) and was euthanised on day 5 after intervention. 

### 3.1. Relevant Timing 

The duration of induction, meaning the time from turning on the isoflurane vaporiser to the loss of rightening reflex, was 311 (253–330) s.

The time intervals from the subcutaneous injection of the study drugs to skin incision were 19.4 (18–29) min, to first whisker stimulation 34.9 (33–45) min, to craniotomy 94.1 (88–104) min, and to stereotaxic vector injection 99.2 (93–109) min. Further, the duration from local infiltration of the incision line to skin incision was 6.9 (2–10) min and to craniotomy 81.9 (73–87) min.

### 3.2. Variables throughout Anaesthesia

Mean body temperature was maintained at 36.4 (range 34.4–36.7) °C. 

Inspired oxygen fraction (F_i_O_2_) ranged from 0.93–0.98.

The MouseOx^®^ Plus showed measurements in thirty-six mice and did not work in five mice (one group C, one group MD, and three group CD mice). The arterial oxygen saturation (SpO_2_) was higher than 95% throughout anaesthesia in 30 mice, partially lower than 95 but higher than 90% in two mice, and partially lower than 90% in four mice. Two of them were in group MD, one in group C and one in group CD.

Heart rates of male and female mice in each group were compared by taking average rates from each animal in 5 min bins throughout the experiment. Animals who received dexmedetomidine had lower heart rates with less variability over the course of the experiment (*p* < 0.0001; two-way ANOVA with Tukey’s post hoc modification; Figure 3a). There was no difference in heart rate at time matched points for groups M and C (*p* > 0.1), or MD and CD (*p* > 0.1; two-way ANOVA with Tukey’s post hoc modification). Measurement of breathing rates throughout the experiment revealed no difference between groups (*p* > 0.1; two-way ANOVA with Tukey’s post hoc modification).

The mean end-tidal isoflurane concentration required by each mouse throughout the experiment was compared and measured in 5 min bins. Dexmedetomidine-treated mice had lower isoflurane requirements than non-dexmedetomidine-treated mice from minutes 55–110 (*p* < 0.01; two-way ANOVA with Tukey’s post hoc modification; Figure 3b). No differences were seen between groups M and C (*p* > 0.1), or MD and CD (*p* > 0.1; Two-way ANOVA with Tukey’s post hoc modification). The mean end-tidal isoflurane concentration for all mice throughout the experiment was 1.8 (0.9–2.4)% *v*/*v* (volume percent).

### 3.3. Variables before and after Defined Stimuli

The cardiorespiratory variables of mice were compared during four surgical and experimental stimuli (skin incision, whisker stimulation, craniotomy, and stereotaxic injection), see Figure 4. Variables were normalised to 2 min averages prior to the start of the stimuli and then compared. Significant increases for groups C and M were identified in both HR (C: *p* < 0.0001; D: *p* < 0.0001; two-way ANOVA with Tukey’s post hoc modification; Figure 4a) and breathing rate (C: *p* < 0.0001; D: *p* < 0.0001; two-way ANOVA with Tukey’s post hoc modification; Figure 4b) during skin incision, but no differences from normalised mean rates for dexmedetomidine-treated mice (*p* > 0.05).

Similarly, mice in groups C and M showed significant increases in heart rate (C: *p* < 0.0001; D: *p* < 0.0001; two-way ANOVA with Tukey’s post hoc modification; Figure 4c) and breathing rates (C: *p* < 0.0001; D: *p* < 0.0001; two-way ANOVA with Tukey’s post hoc modification; Figure 4d) during whisker stimulation. There were no differences from normalised mean rates for dexmedetomidine-treated mice (*p* > 0.05).

Interestingly, no differences in variables were found during craniotomy in any group (heart rate: *p* > 0.1; two-way ANOVA with Tukey’s post hoc modification; Figure 4e; breathing rate: *p* > 0.1; two-way ANOVA with Tukey’s post hoc modification; Figure 4f). 

Furthermore, variables during stereotaxic injection of viral vectors were compared. No differences in heart rates were found in any of the treatment groups (*p* > 0.1; two-way ANOVA with Tukey’s post hoc modification; Figure 4g). A significant increase in breathing rates was detected for group C only (*p* < 0.0001; D: *p* < 0.0001; two-way ANOVA with Tukey’s post hoc modification), with no differences in any of the other groups (*p* > 0.1; two-way ANOVA with Tukey’s post hoc modification; Figure 4h).

### 3.4. Postoperative Treatment

One mouse (group C) received rescue analgesia (buprenorphine 0.1 mg kg^−1^ subcutaneously) twice and five other mice once during the postoperative period (one in group CD, one in group MD, and three in group M). Furthermore, twenty-eight mice (five in group C, nine in group CD, six in group M, eight in group MD) received supplementary sugar-rich food because they showed continued weight loss of maximally 5–10% between day one and day four (one mouse four times, one mouse three times, ten mice twice, and sixteen mice once). A fluid bolus of 10 mL kg^−1^ was administered to one mouse (group C).

## 4. Discussion

In order to identify an appropriate, intervention-adapted anaesthetic regimen and to improve the intraoperative analgesia in mice undergoing craniotomy, this study examined four drug combinations and their respective effects regarding analgesia and cardiovascular and anaesthetic stability. It was found that the administration of dexmedetomidine resulted in significant reductions of heart and breathing rates at two of the four assessed stimuli (skin incision and whisker stimulation). Besides this, the mice treated with dexmedetomidine not only showed lower heart rates with less variability throughout general anaesthesia but also a reduction in volatile anaesthetic agent requirements. We further detected that there was a difference between meloxicam- and carprofen-treated mice during stereotaxic injection, when the breathing rate in mice treated with carprofen was higher. 

Dexmedetomidine, a highly α_2_/α_1_-selective α_2_-adrenergic agonist in the central and peripheral nervous systems, binds and stimulates α_2_-adrenoreceptors, providing sedation, analgesia, anxiolysis, muscle relaxation, and a reduction in MAC [24,25]. Further, it has been shown that tissue oxygen consumption is reduced, and therefore dexmedetomidine acts as a neuroprotective drug [26,27,28], whilst providing haemodynamic stability [16,28]. As in humans, it is used during and following neurosurgery [11,17]. Therefore, we reasoned that it could also be beneficial for cranial surgery in mice, as isoflurane used alone for single-drug anaesthesia causes cardiovascular instability [29] and cognitive impairment [30,31] in rodents. Cardiovascular side effects, such as reduced heart rate, mean arterial blood pressure, and myocardial contractility, have been reported in a dose-dependent manner with increasing isoflurane levels [29]. It was found that an isoflurane dose level of 1.5% *v*/*v* (volume percent) provides stable mean arterial blood pressure and heart rate values in mice compared to lower or higher isoflurane dosages [32]. These values were collected and analysed in animals not exposed to noxious stimuli, indicating that isoflurane dosages of 1.5% *v*/*v* would not be sufficient for mice undergoing invasive surgical procedures. This is confirmed by our study, where mice showed isoflurane requirements of 1.8 (0.9–2.4)% *v*/*v* under the influence of additional drugs with analgesic and sedative properties. Assuming that for invasive surgical procedures with isoflurane alone, the dose requirement would be even higher, it is likely that cardiovascular function would deteriorate with an increasing risk of morbidity and mortality. The influence of dexmedetomidine on the minimum alveolar concentration (MAC) of isoflurane has not yet been investigated in mice, but it is well-known that this drug reduces MAC in rats and other species [12,13,14]. 

Nociception and pain can cause distinct cardiovascular effects, like alterations in heart rate and heart rate variability [33]. Under general anaesthesia with inadequate analgesia, tachycardia, hypertension, and tachypnoea can be seen in clinical patients undergoing surgery [34]. A reduction in such reactions to surgery is indicative of adequate analgesia blocking the sympathetic and parasympathetic reactions. In our study, significant increases in both heart rate and breathing rate for non-dexmedetomidine-treated animals during skin incision and whisker stimulation, which are both strong stimuli for mice, were recorded [35,36]. Also, during the main part of the surgery, 55–105 min after study drug administration, lower isoflurane requirements were observed. It can be concluded that nociception and MAC were reduced by dexmedetomidine. This is in agreement with other studies that found a dose-dependent analgesic effect for dexmedetomidine up to 120 min in mice subjected to a chemical acetic acid-induced writhing assay and a thermal tail-flick test [21,23], although its analgesic properties have not been reported in mice undergoing surgery yet. In the present study, the administration of a non-steroidal anti-inflammatory drug (NSAID) and the local infiltration with lidocaine did not provide sufficient analgesia for skin incision and whisker stimulation, as groups C and M showed a reaction to surgery. Following local infiltration of the incision line using lidocaine 2 mg kg^−1^ the skin incision was conducted after 6.9 (2–10) minutes (Figure 1a). This could have been too early considering the reported onset of action of 5–15 min for the local administration of lidocaine [37]. The time interval from subcutaneous administration of the analgesia-related study drugs to skin incision was 19.4 (18–29) minutes and the first whisker stimulation was performed at 34.9 (33–45) minutes after injection, followed by three to four whisker stimulations (Figure 1 a). Pharmacokinetic research determined the time to reach maximum concentration (T_max_) of meloxicam to be 1–2 h for dosages of 1.6–5 mg kg^−1^ [38,39] and of carprofen to be 2 h for a dosage of 5 mg kg^−1^ [39,40]. Therapeutic plasma levels of both NSAIDs are not known yet for mice but can be rated from other species where the cyclooxygenase inhibiting effect was proven with 20–24 µg mL^−1^ of carprofen [41] and 390–911 µg mL^−1^ for meloxicam [42,43,44]. Whereas carprofen maintained plasma levels higher than the estimated therapeutic level for 12 h, meloxicam concentrations dropped below the level after 4 h [39]. Including the T_max_ values mentioned before, these data indicate a therapeutic plasma level for both drugs during craniotomy and vector injection. Nevertheless, the breathing rate in group C was higher during stereotaxic viral vector injection. Possibly, the dose rates used were not equipotent. The dose used for both NSAIDs (5 mg kg^−1^), is common [5] and suggested by reference literature [45,46] and was therefore chosen for the present study. However, efficacy studies showed that dosages of 5 mg kg^−1^ of carprofen and 1 mg kg^−1^ of meloxicam provide some analgesic effects [39], whereas other studies found that dosages under 20 mg kg^−1^ of both, meloxicam and carprofen, do not have analgesic effects in mice [47,48]. It remains to be investigated whether higher doses of meloxicam and carprofen are sufficient to provide analgesia, resulting in stable cardiopulmonary function during craniotomy.

During craniotomy, there were no significant differences in heart rates and breathing rates in any group, suggesting that by then analgesia was sufficient (Figure 4e,f). The procedure of drilling the skull for the craniotomy is thought to be a strong stimulus, although it does not really seem to cause more nociception compared to other stimuli such as skin incisions in people [49]. The reason for less nociception during craniotomy in the present study is probably related to the lidocaine applied locally. In rats, it was shown that lidocaine directly penetrates the underlying tissue levels up to a depth of 1 cm with concentrations of 0.25% in the first 2 h after application [50]. Considering that in the current study the time interval from administration of the respective analgesic drug combination to craniotomy was 94.1 (88–104) min and to vector injection was 99.2 (93–109) min (Figure 1a), the penetration of lidocaine with a concentration of 2% up to the level of the pericranium and the sustained analgesic effect are reasonable. 

Dexmedetomidine has a minimal impact on the respiratory control systems in most species, including rodents [24,25,51,52]. Accordingly, we found no significant differences in breathing rates between the groups with and without dexmedetomidine. However, the dose level of isoflurane was adjusted according to breathing rate, and therefore a potential difference in breathing rate between the groups was masked. Importantly, however, ventilation was sufficient to maintain oxygenation well in all but four mice, one in group C, one in group CD, and two in group MD. Considering that an arterial oxygen saturation lower than 95% is defined as hypoxaemia and lower than 90% as severe hypoxaemia [53], these four mice with saturation values partially lower than 90% most probably suffered hypoxaemia, if measurements were accurate. There are no reports investigating the reliability of the MouseOx^®^ Plus. For other pulse oximeters, it has been shown that poor peripheral perfusion caused by vasoconstriction, a potential side effect of dexmedetomidine, can lead to erroneously low SpO_2_ values [54]. As three of the four mice with values lower than 90% were treated with dexmedetomidine, vasoconstriction cannot be the only reason. In our study, there were also five mice for which the pulse oximeter did not provide any measurements. As these mice were either dexmedetomidine- or non-dexmedetomidine-treated, vasoconstriction seems not to be the only influence regarding the reliability of the measurement. It is known that skin pigmentation or fur can interfere with pulse oximetry readings as well [55]. To reduce the risk of this factor, the experimental setup of our study included shaving of the medial and lateral aspects of the thigh, which was used for pulse oximetry. To reliably judge oxygenation status in all mice, we would have had to draw arterial blood samples for blood gas analyses [56], a challenging invasive procedure that cannot be used to constantly monitor the oxygenation status of a mouse; therefore, we used the less precise method, which is prone to a certain number of measurement problems. 

The present study has several limitations. Firstly, the relatively low number of animals and the lack of statistical power analysis. There was no preliminary data to perform a statistical power analysis, but compared to other efficacy studies using 6–21 animals [38,57,58] per treatment group our, group size of ten mice seems to be reasonable. Another limitation is the inconsistency of measurements via the MouseOx^®^ Plus and capnography due to the small body size of mice, which impeded a continuous and stable observation of all parameters. In order not to bias the results, the data analysis was adapted by taking average rates from each animal in 5 min bins for investigation of anaesthesia stability and required isoflurane concentrations. To elucidate the analgesic effect of the different drug combinations during the four stimuli, variables were normalized to 2 min averages prior to the start of the stimuli and then compared. Dexmedetomidine levels were not assessed and correlated with the observed effects. Other useful biomarkers to assess surgical-anaesthetic stress [59] were not measured.

The current study’s aim was to investigate efficient, reliable, and practicable anaesthesia and analgesia regimens for optimisation of intraoperative analgesia in mice undergoing craniotomy. Reasons against the use of opioids in mice are their strong systemic side effects, such as bradycardia and respiratory depression [60] in general, and the behavioural side effects of buprenorphine in particular, which include hyperlocomotion, tail extensions, circling and tiptoe gait, as well as pica behaviour [58]. 

## 5. Conclusions

In the current study, the administration of dexmedetomidine increased intraoperative cardiovascular stability, provided an analgesic effect, and less isoflurane was necessary to maintain anaesthesia, whereas the analgesic effect of carprofen and meloxicam (5 mg kg^−1^ each) combined with local anaesthesia using lidocaine was not adequate. In conclusion, the anaesthetic regimen with isoflurane, dexmedetomidine, NSAIDs, and locally applied lidocaine may provide welfare refinement to mice undergoing craniotomies.

## Figures and Tables

**Figure 1 animals-14-00913-f001:**
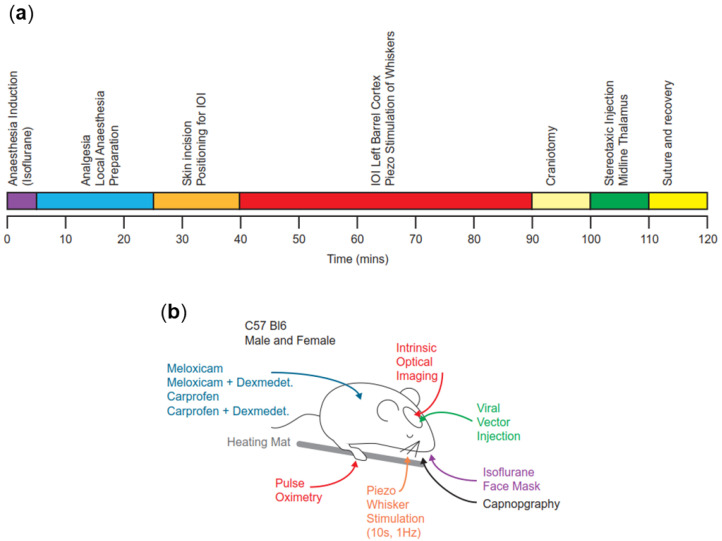
Experimental setup. (**a**) Experimental timeline for all procedures under isoflurane general anaesthesia. At 120 min, animals were placed in a clean cage for postoperative monitoring. (**b**) Schematic of experimental procedures carried out under general anaesthesia. Note that procedures were carried out in sequence as per panel a, and not simultaneously.

**Figure 2 animals-14-00913-f002:**
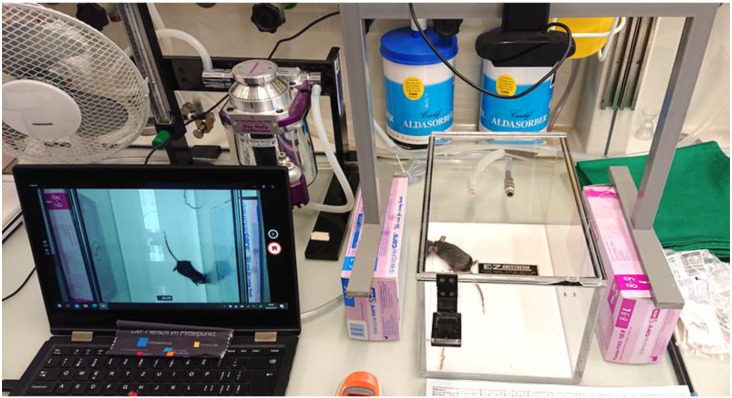
Experimental setup for anaesthesia induction. Oxygen flowmeter with an isoflurane vaporiser, set to 3.0% in 1.0 L min^−1^ (reduced to 2.0% after 3 min), is tubed to a plexiglass chamber providing reliable anaesthesia induction in the mouse.

**Figure 3 animals-14-00913-f003:**
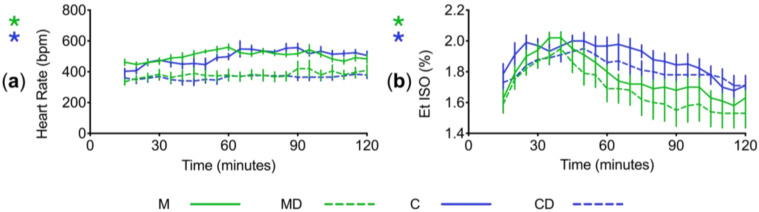
Intraoperative variables throughout anaesthesia. (**a**) Mean heart rate ± S.E.M. recorded at 5 min intervals throughout the experiment for all four treatment groups. Dexmedetomidine resulted in a reduction in heart rate (*p* < 0.0001); whereas there was no difference between groups M and C (*p* > 0.1), or MD and CD (*p* > 0.1; two-way ANOVA with Tukey’s post hoc modification). Note that measurements begin at 15 min when instrumentation occurred; (**b**) mean end-tidal isoflurane concentration ± S.E.M. recorded at 5 min intervals throughout the experiment. Significant differences were found between all groups between 55 and 110 min (*p* < 0.01; two-way ANOVA with Tukey’s post hoc modification). ***** = significant (*p* < 0.05).

**Figure 4 animals-14-00913-f004:**
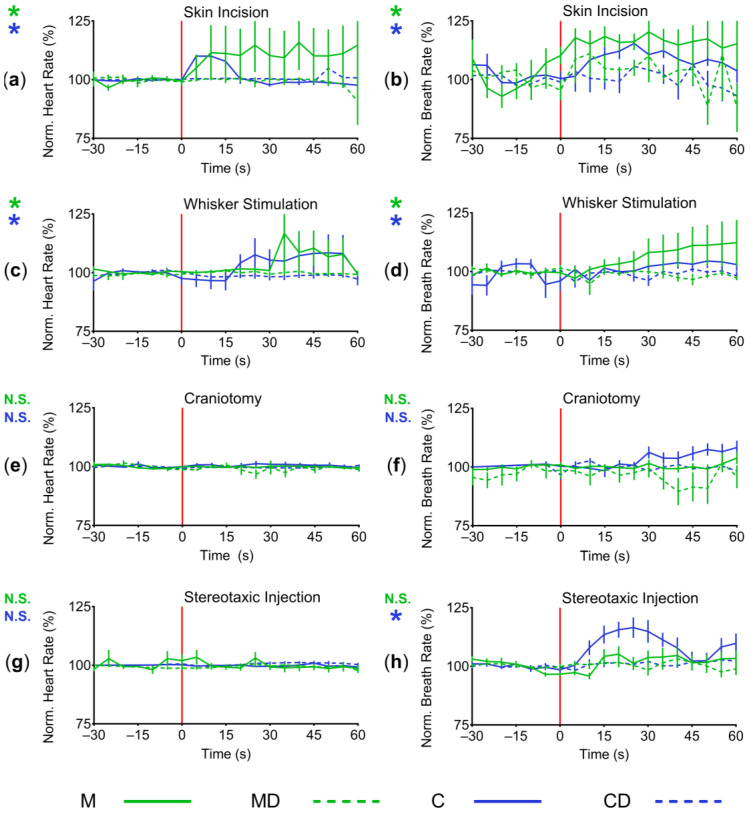
Intraoperative variables, specific stimuli. (**a**) Mean heart rate ± S.E.M. from 30 s before to 60 s after initial skin incision. Skin incision is marked by a red vertical line at t = 0 s. There was a significant increase in heart rate in groups M and C (*p* < 0.0001; two-way ANOVA with Tukey’s post hoc modification); (**b**) Mean breathing rate ± S.E.M. from 30 s before to 60 s after initial skin incision. Skin incision is marked by a red vertical line at t = 0 s. There was a significant increase in breathing rate in groups M and C (*p* < 0.0001; two-way ANOVA with Tukey’s post hoc modification); (**c**) Mean heart rate ± S.E.M. from 30 s before to 60 s after whisker stimulation. Whisker stimulation is marked by a red vertical line at t = 0 s. There was a significant increase in heart rate in groups M and C (*p* < 0.0001; two-way ANOVA with Tukey’s post hoc modification); (**d**) Mean breathing rate ± S.E.M. from 30 s before to 60 s after whisker stimulation. Whisker stimulation is marked by a red vertical line at t = 0 s. There was a significant increase in breathing rate in group M (*p* < 0.0001; two-way ANOVA with Tukey’s post hoc modification); (**e**) Mean heart rate ± S.E.M. from 30 s before to 60 s after craniotomy. Craniotomy is marked by a red vertical line at t = 0 s. No significant differences were found in any group (*p* > 0.1; two-way ANOVA with Tukey’s post hoc modification); (**f**) Mean breathing rate ± S.E.M. from 30 s before to 60 s after craniotomy. Craniotomy is marked by a red vertical line at t = 0 s. No significant differences were found in any group (*p* > 0.1; two-way ANOVA with Tukey’s post hoc modification); (**g**) Mean heart rate ± S.E.M. from 30 s before to 60 s after the start of stereotaxic injection. Start of stereotaxic injection is marked by a red vertical line at t = 0 s. No significant differences were found in any group (*p* > 0.1; two-way ANOVA with Tukey’s post hoc modification); (**h**) Mean breathing rate ± S.E.M. from 30 s before to 60 s after start of stereotaxic injection. Start of stereotaxic injection is marked by a red vertical line at t = 0 s. There was a significant increase in heart rate in group C (*p* < 0.0001; two-way ANOVA with Tukey’s post hoc modification). * = significant (*p* < 0.05); N.S. = not significant.

## Data Availability

The data presented in this study are available on request from the corresponding author.

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
