# Peer review of "Antinociceptive and Cardiorespiratory Effects of a Single Dose of Dexmedetomidine in Laboratory Mice Subjected to Craniotomy under General Anaesthesia with Isoflurane and Carprofen or Meloxicam"

_animals, 2024, doi:10.3390/ani14060913_

Round 1

Reviewer 1 Report

Comments and Suggestions for Authors

The article by Schiele et al. explains in great detail a study in which the usefulness of the use of dexmetomidine for anti-nociception during cranial surgery is evaluated.

Its content is extremely interesting, as it provides very useful information for researchers performing surgical models in mice (not only for neurosurgery).

I consider that this type of studies, with a design as correct as the one presented, are as scarce as they are necessary and therefore I recommend its publication.

COMMENTS

- The writing of the Introduction and Discussion sections should be revised, as the thread is sometimes not coherent. For example, the information of the last paragraphs should be inserted in previous paragraphs, as it is partly repetitive.

- The Results section should include data on ALL the variables recorded. Data on body temperature and oxygen saturation are missing, although they are mentioned in the Discussion.

- For a quick interpretation of the graphs, the variables with statistically significant differences may be marked with an asterisk, and the rest with an "N.S." (not significant). (not significant).

Reviewer 2 Report

Comments and Suggestions for Authors

The group sought to determine the benefits of the alpha agonist dexmedetomidine when applied in concert with isfluorane general anesthesia, lidocaine local anesthesia, and NSAIDs during craniotomy. Benefits were determined by quantifying outcome measures such as heart rate, respiratory rate, and isofluorane concentration. These outcome measures in turn served as indicators for experienced pain during noxious stimuli.  

The measured reduction in pain indicators is relevant to the field. The specific gap is related to the use of alpha agonists (i.e. dexmedetomidine), lidocaine, and NSAIDs in the context of murine craniotomy pain mitigation.

This adds a specific examination of the benefits of multiple drugs, namely dexmedetomidine, during murine craniotomy.

It would be difficult if not impossible to ethically introduce a true control in this case. No surgery can be ethically performed without pharmaceutical intervention. The closest thing to an ethical control could be a craniotomy performed under isoflurane induced general anesthesia and nothing else.

The conclusions generally agree with the results presented. Findings do not appear to be overstated - this is not a truly seminal paper. Instead it contributes to the important body of literature that examines the problem of pain management in laboratory animals. At line 493 I think a brief description of what the authors mean by "isofluorane sparing effect" would be helpful. Does this just mean less gas is needed during the course of the surgery? I would be curious to see a cost-comparison of the drugs used since the addition of dexmedetomidine may represent a significant cost in certain cases.

References appear to be appropriate

Figure 1 is very good and is helpful for reviewing the chronology of surgery. Figure 2 could benefit from some kind of indicator that denotes timepoints at which statistically significant differences were measured. The same goes for data presented in Figure 3.

There is a typo at line 401 - "recoded" should likely read "recorded".  A brief explanation of the purpose of whisker stimulation would be useful as well. It is stated that it is a stimulus for mice but is it a true analogue for pain?

Comments on the Quality of English Language

English quality is generally fine - just fix "data" use. The word data is plural. 

Reviewer 3 Report

Comments and Suggestions for Authors

Dexmedetomidine is a drug that has found greater use in human and veterinary medicine. Scientific evidence on dexmedetomidine has increased exponentially, so its clinical use is no longer limited to sedative properties, but perioperative analgesic effects have also been documented in opioid-free anesthetic plans, and even as an adjuvant in loco-regional anesthesia. Other pharmacological characteristics that have also been the subject of study are the cardiovascular effects since it is a drug that can play an important role in reducing oxygen demand, especially of the heart, a mechanism that is beneficial in critically ill patients. Another effect that has emerged in recent years is the vascular protective role, where dexmedetomidine has found wide use in critically ill, traumatized, and septic patients. The present study aims to evaluate the antinociceptive and cardiorespiratory effects of dexmedetomidine in laboratory mice subjected to craniotomy under general anesthesia with isoflurane and carprofen or meloxicam, which may be its main strength because rodents are one of the model’s studies for this surgery.

However, some points must be addressed to achieve publication quality. I have left some comments hoping that they can help the authors.

General comments

The title of the manuscript does not correctly establish its content, since it does not refer to the central topic of the research, so I suggest the authors make the following modification:

“Antinociceptive and cardiorespiratory effects of a single dose of dexmedetomidine in laboratory mice subjected to craniotomy under general anesthesia with isoflurane and carprofen or meloxicam”

L22-23: The study aim is confusing, please rephrase.

L28-29: I suggest the authors rewrite these lines, presenting the results of said variables in mean and standard deviation with the P value in parentheses so that the reader observes the statistical significance.

L29: what were the harmful stimuli? Please clarify.

L30: How much was the anesthetic requirement of isoflurane reduced? Write this value in percentage.

L71-75: These aspects should be discussed in greater depth in the discussion.

L97-101: I suggest the authors write the aims more clearly. Moreover, add the study hypothesis.

L105: please, indicate the type of study; prospective, randomized, blind…

L112: What was the statistical method by which the sample size was determined? Please clarify

L112-116: the inclusion and exclusion criteria should be explained more clearly.

L139: please add an image of the induction chamber used.

L140 and 198: was the oxygen used 100%?

L270-271: please add on these lines the anesthesia time, surgery time, and anesthetic recovery time for each study group. In addition to these data, indicate how many animals had analgesic rescue, and how many were excluded due to hyperaggressive handling, immobility, non-healing skin incisions, or pain.

L284: what percentage reduction in ETISO was generated in the groups treated with dexmedetomidine? And is this value adjusted to barometric MAC? Please clarify.

L299: although they are indicated in Figure 3, please clarify in this line what the surgical and experimental stimuli were.

L359: Intraoperative monitoring of ETCO2, FiO2, and temperature is mentioned in the methodology. I suggest the authors include the results obtained in these variables.

L360: In general, the discussion should deeper into the cardiorespiratory, analgesic, and neuroprotective effects of dexmedetomidine in rodents.

L365-366: which stimuli?

L398: inadequate analgesia not only generates hemodynamic instability but there are more associated changes. I suggest the authors review this document to complement their discussion:

10.14202/vetworld.2021.393-404

L441: These findings are also related to the ETISO used. 

L473-484: Please discuss other limitations of your study, for example, lack of postoperative pain evaluation using the rat grimace scale to assess pain, lack of correlation of cardiorespiratory changes with pharmacokinetic parameters of dexmedetomidine, measurement of other biomarkers such as lactate, cortisol, glucose, C-reactive protein, malondialdehyde, among others.

Comments on the Quality of English Language

Minor editing of English language required

Reviewer 4 Report

Comments and Suggestions for Authors

Line 56 – place a comma after ‘However’.

Line 60-61 – did you mean to make that statement for ‘craniotomies performed under isoflurane anesthesia in laboratory rodents’?

Line 62-63 – grammatical error about placement of commas. It should be – “In humans undergoing craniotomy, intraoperative constant rate infusion of dexmedetomidine reduced postoperative pain, the consumption of other analgesic agents, and frequently ….”.

Line 64-65 – review grammar and rephrase.

Line 72 – check grammar. Replacing ‘has’ with ‘is’ will probably be better. And it should be “..but also ‘has’ analgesic….”

Line 76, 84, 98, 99 – needs a comma.

Line 151 – states application of Vitamin A eye cream in eyes. That cream is used for application on skin around eyes and not in the eyes. Usually, artificial tears or other ocular lubricant is used for such application in eyes. Please check what was used (provide product info in parentheses) and correct this.

Line 154 – states “diluted to a total volume of 5mL kg-1”. That is an incorrect statement. The sentence should be corrected to reflect that the drug dilutions were made such that injectate volume for each drug based on dose calculation was 5ml/kg.

Line 155 and 157 – the IU units are listed in parentheses. Those don’t belong there. Check and remove those.

Line 148-149 states use of a non-feedback heating mat (Tonkey Electrical Technology Co. Ltd, China), which is different than description in line 174, which states ‘..feedback to a heating mat (Homeothermic Monitoring System)..’. Is the statement in line 148-149 for the heating mat placed under induction box? The statement “..and positioned with the nose in a gas delivery mask for…” in line 150 makes it confusing if the non-feedback heating mat is used where face mask is used and that is described in line 174 as a different type heating mat.

Line 176 – states “sterile swab for saving heat”. That seems to be a mistake and should likely be ‘sterile drape for saving heat’. Please check and correct.

Line 189 – states use of Meliseptol for surgery site preparation. That product is for use as disinfectant for surfaces and devices, not for surgery site preparation.

Line 170 describes use of sampling tube placed directly in front of the mice nares. However, line 195 describes use of face mask for isoflurane delivery. Please include more details to clarify how/where the sampling tube was placed.

Line 240 – “After another two days of surveillance, animals were transferred to the EIVC cages and brought back to standard husbandry..”. What type of caging and location was used over these 7 days (5 days NSAID and additional 2 days) post-surgery? Were they housed individually in open top cages as mentioned in line 236 for anesthetic recovery?

Line 248-250 – needs grammar check and correction.

Results section – can you add subtitles at beginning of each paragraph.

Line 401 – typo error – ‘recoded’ should be corrected to ‘recorded’.

Comments on the Quality of English Language

Lot of grammar and punctuation corrections are needed. Please consult with a scientific English language specialist.

Round 2

Reviewer 3 Report

Comments and Suggestions for Authors

I thank the authors for considering my comments and observations from the first revision of your manuscript. It seems to me that the article has improved substantially, so I am convinced that if this material is published, readers will have an important contribution to favor the development of veterinary medicine, particularly in the field of surgery and anesthesia of laboratory animals.

However, in the new manuscript, I have found some aspects that require the attention of the authors before publishing the article.

L32: review the percentage of anesthetic reduction, in this reviewer's experience dexmedetomidine reduces ETISO by 17-53% when used as a single bolus.

L74 and 79: in both lines, a hypothesis is proposed, please homologate it to only one.

L117: The image from the induction chamber seems correct to me. Please add an explanation of the figure to be added to the manuscript.

L241: I have calculated the sample size for your study. I suggest the authors include the following paragraph:

“The sample size was calculated using G*Power 3.1.9.7 (Heinrich-Heine-Universität Düsseldorf, Düsseldorf, Germany). The total sample size was 24 animals, considering an error probability of 0.05, confidence 95% level, power (1-b probability of error) of 0.90, and correction between repeated measures of 0.5 for four experimental groups”.

L272: what device was used for temperature maintenance? Please clarify in the section on methodology.

L273: please indicate the capnography values ​​recorded in the monitoring.
